# The Never-Ending History of Octreotide in Thymic Tumors: A Vintage or A Contemporary Drug?

**DOI:** 10.3390/cancers14030774

**Published:** 2022-02-02

**Authors:** Liliana Montella, Margaret Ottaviano, Rocco Morra, Erica Pietroluongo, Pietro De Placido, Marianna Tortora, Chiara Sorrentino, Gaetano Facchini, Sabino De Placido, Mario Giuliano, Giovannella Palmieri

**Affiliations:** 1ASL NA2 NORD, Oncology Operative Unit, “Santa Maria delle Grazie” Hospital, 80078 Pozzuoli, Italy; gaetano.facchini@aslnapoli2nord.it; 2Oncology Unit, Ospedale del Mare, 80147 Naples, Italy; margaretottaviano@gmail.com; 3CRCTR Coordinating Rare Tumors Reference Center of Campania Region, 80131 Naples, Italy; sabino.deplacido@unina.it (S.D.P.); m.giuliano@unina.it (M.G.); 4Department of Clinical Medicine and Surgery, Università degli Studi di Napoli “Federico II”, 80131 Naples, Italy; rocco.mor4@gmail.com (R.M.); erica.pietroluongo@gmail.com (E.P.); pietrodep91@gmail.com (P.D.P.); marian.tortora@gmail.com (M.T.); chiarasorrentino87@gmail.com (C.S.)

**Keywords:** thymic epithelial tumors, thymoma, thymic carcinoma, targeted therapy, somatostatin, octreotide, prednisone

## Abstract

**Simple Summary:**

Thymic epithelial tumors are rare tumors frequently associated with paraneoplastic syndromes, the most common being myasthenia gravis and pure red cell aplasia. While patients with limited-stage cancer can often undergo resolutive surgery, advanced surgically unresectable and metastatic tumors can be refractory to first-line platinum-based treatment and represent a medical challenge. Somatostatin receptor expression was documented in thymic tumors both in vivo and in vitro and represents the rationale for therapeutic use. Despite single-case reports and three single-arm phase II studies, as well as the inclusion of somatostatin analogs in National Comprehensive Cancer Network guidelines, the role of these drugs in thymic epithelial tumors is still rather undefined.

**Abstract:**

Thymic epithelial tumors are rare tumors usually presenting as a mass located in the anterior mediastinum and/or with symptoms deriving from associated paraneoplastic syndromes. Unresectable platinum-refractory tumors are often treated with alternative regimens, including chemotherapeutic agents as well as chemo-free regimens. The most popular unconventional therapy is represented by the somatostatin analog octreotide, which can be used alone or with prednisone. The in vivo expression of somatostatin receptors documented by imaging with indium-labeled octreotide or gallium-68 Dotapeptides, the successful use of octreotide and prednisone in a chemo-refractory patient, and, thereafter, the experiences from a case series have enforced the idea that this treatment merits consideration—as proved by its inclusion in the National Comprehensive Cancer Network guidelines. In the present review, we analyze the preclinical basis for the therapeutic use of somatostatin and prednisone in refractory thymic tumors and discuss the available studies looking at future perspectives.

## 1. Introduction

Thymic epithelial tumors (TETs) are rare tumors (1.5 cases/million) [1] usually presenting as a mediastinal mass and/or with symptoms derived from associated paraneoplastic syndromes. The 2021 WHO Classification differentiates TETs into thymomas (subtyped into A, AB, B1, B2, and B3), thymic carcinomas, and thymic neuroendocrine tumors. A criterion to differentiate thymomas is the lack of immature T cells—typical of A thymomas—versus an increased number of immature T cells—a feature of AB, B1, B2, and B3 thymomas. This morphology matches the association with related immune disorders, especially paraneoplastic myasthenia, which occurs predominantly in B thymomas. The classification from A to thymic carcinoma tumor reflects the transition from indolent to more aggressive types and a difference in the 5-year survival rate, which is 90% in thymomas and around 55% in thymic carcinoma [1]. While complete resection may be resolutive in indolent and limited-stage tumors, advanced and recurrent tumors remain a medical challenge. After the failure of platinum-based regimens, several treatments can be considered [1]. However, it is increasingly difficult to control the disease. The treatments for refractory tumors include alternative chemotherapeutic regimens—such as capecitabine plus gemcitabine [2,3]—immune therapies [4], and targeted therapies [5]. In a series of refractory TETs with a majority of thymomas, a significant rate of disease control (responses plus disease stabilization: 80%) and a median progression-free survival (PFS) reaching 11 months were obtained with capecitabine plus gemcitabine. This was one of the last chemotherapeutic regimens proposed before the advent of the precision medicine era in TETs. In fact, since 2010, progress in genetics has contributed to differentiating thymomas from thymic carcinomas, each one with a distinctive pattern recognized in the 2021 WHO Classification [6].

The genomic analysis defined a thymoma-specific mutated oncogene General Transcription Factor IIi (GTF2I) and a prevalent pattern of mutations in HRAS, NRAS, and TP53 [7]. TETs show the lowest average tumor mutation burden (TMB) among adult cancers, and actionable genes have not been detected [8]. With regard to biomarkers as being possibly predictive of a response to immunotherapy, some observations indicate that this therapy may be a promising choice at least in some types of TETs, and that the potential benefits outweigh the risks of unwanted immune-related adverse events. Immune cell infiltration and high expression of PD-L1 on tumor cells are in favor of immunotherapy in TET [9]. Recently, another intriguing finding has been added to support the use of immune checkpoint inhibitors in TETs. High-TMB TETs were significantly associated with poor prognostic features, such as a later stage, more advanced pathological type, older age, and ultimately a poor prognosis, as compared with low-TMB TETs [10]. This latter distinction may configure a tailored use of ICIs.

Targeted therapies, the other front of precision medicine therapies, have also been tested in refractory TETs. In this setting, such strategies find a biological prerequisite in the change occurring in advanced pretreated tumors, thereby showing an increasing number of somatic mutations (more than 60% in thymic carcinomas) as compared to untreated ones [8]. The genetic differences between thymomas and thymic carcinomas may also help define different possible targets [11]. For example, c-Kit overexpression was found in thymic carcinomas and not in thymomas [12]. However, the clinical use of imatinib, which notoriously is a tyrosine kinase inhibitor targeting kit, reported contrasting results [13,14,15]

The most relevant results from targeted therapies in TETs were achieved by everolimus and sunitinib. Everolimus is an inhibitor of the serine–threonine kinase mammalian target of rapamycin (mTOR), a key component of the PI3K/AKT/mTOR intracellular axis. This signaling pathway rules cell growth, proliferation, and angiogenesis. A phase II trial involving 51 patients reported a durable disease control rate (88%) in pretreated TETs [16]. A significant response rate was also achieved by sunitinib in two studies: an overall response as high as 63% in one series with 28 patients [17] and 26% of partial responses in thymic carcinomas were reported [18]. Recently, promising results were achieved in the phase 2 multicenter trial with the multitargeted tyrosine kinase inhibitor lenvatinib in a pretreated population [19].

The chance to control the disease with drugs interfering with key targets and minimize the toxic effects and immune suppression of cancer with typical defective immunotolerance, and a predisposition to autoimmune disorders, inspired our earlier studies concerning somatostatin analogs. In this review, we revised the topic and discussed future conceivable perspectives.

## 2. Preclinical Rationales for Somatostatin Analog Use in TETs

### 2.1. Somatostatin and Somatostatin Receptors

Somatostatin, a small peptide with predominantly inhibitory effects, though present throughout the body, is mainly found in the gastrointestinal tract and the nervous system. Two bioactive peptides, somatostatin-14 and somatostatin-28 are generated from the cleavage of the large precursor molecule and induce several cell functions through interaction with specific membrane receptors and triggered signaling. Five subtypes of receptors (sst_1–5_) have been identified, all with a high affinity for somatostatin-14 and 28 [20]. These receptors present a differential tissue distribution, regulation, and binding to different somatostatin analogs.

The sst_2_ gene product is alternatively spliced to encode sst_2A_ and sst_2B_ with a different carboxy-terminal sequence. The somatostatin receptor family belongs to G-protein-coupled receptors, all five subtypes being functionally linked to adenylate cyclase through a coupling mechanism involving the guanine-nucleotide-binding (G) protein. The binding of somatostatin to the receptor reduces the intracellular levels of cyclic AMP and ionized calcium and increases tyrosine phosphatase activity.

More than one somatostatin receptor subtype with a predominant expression of sst_2_ is usually found in most tumors. The analysis of the functional role of each subtype is unfeasible because of the simultaneous expression of multiple subtypes by the same tissue. Moreover, the landscape of somatostatin signaling has been enriched by another piece of information recently: receptor subunits can homo- or hetero-dimerize among themselves and with other G-protein-coupled receptors [21].

The physiological activities of somatostatin range from classically known modulation of gastro-enteric-pancreatic cells and endocrine activity to involvement in immune responses and neurotransmission [22]. Somatostatin has inhibitory effects on secretive and proliferative processes, reducing cell proliferation, increasing apoptosis, and inhibiting angiogenesis in most tumor tissues [20]. The antiproliferative effects are either direct or indirect. The direct effects involve the suppression of the synthesis of autocrine/paracrine growth-promoting hormones and growth factors, inhibition of mitogenic signals mediated by growth factors, and induction of apoptosis [22]. Cell growth arrest is induced by several mechanisms, such as upregulation of cyclin-dependent kinase (CDK) inhibitors, regulation of MAP kinases, stimulation of tyrosine phosphatases, inhibition of Na^+^-H^+^ exchanger, and modulation of nitric oxide production [21,22]. Sst_3_, more than other subtypes, appears to be involved in apoptosis. Somatostatin controls tumors indirectly by inhibiting angiogenesis and immune-modulatory effects. The antiangiogenic effect may be at least partially attributed to the reduced secretion of vascular endothelial factor production, as shown in cell cultures [23]. Endothelial cells express sst_2_ and sst_5_ [24]. Several experimental results indicate that somatostatin and its analogs inhibit angiogenesis in vitro and in vivo. Unlike quiescent endothelial cells, proliferating endothelial cells were shown to express sst_2_ and sst_5_. Octreotide and SOM230, which is a somatostatin analog with a high binding affinity for all somatostatin receptor subtypes apart from sst_4_, were able to inhibit proliferating cells in an experimental model in a dose-dependent manner [24].

Somatostatin receptor expression was studied firstly at the mRNA level by in situ hybridization and a reverse transcriptase polymerase chain reaction (RT-PCR), and thereafter by immunohistochemistry performed with polyclonal antibodies specific for each subtype in order to study the protein expression in rat and human tissues. In vivo, somatostatin receptor scintigraphy, also called an octreotide scan, helps visualize sst-expressing tissues. Normal tissues, such as the brain, the pituitary, the thyroid, the pancreas, the spleen, kidneys, blood vessels, the peripheral nervous system, immune cells, and a wide range of malignancies [25,26,27], express ssts—with the greatest expression being by well-differentiated neuroendocrine tumors [28]. Lymphoid cells, especially when activated, also express ssts, [29] thereby enabling the in vivo visualization of inflammatory diseases by somatostatin receptor scintigraphy (SRS). Human peripheral blood B- and T-lymphocytes were shown to selectively express sst_3_, while monocytes are induced to express sst_2A_ upon activation [30]. The first and most common type of imaging based on these receptors comprises 111 In-DTPA (diethylenetriamine pentaacetate)-D-Phe-1-octreotide, which mainly binds to sst_2_ and sst_5_. The second and newest type of SRS uses somatostatin analogs marked by the positron emitter gallium (Ga), including Ga-DOTATOC (DOTA-Tyr3-octreotide), Ga-DOTANOC (1-Nal3-octreotide), and Ga-DOTATATE (DOTA-(Tyr)-octreotate) [31]. Somatostatin receptor imaging can now be performed with positron emission tomography (PET), which represents a quick and high-resolution form of imaging. As well, gallium-68 receptor PET-CT is more specific than an ^111^In-octreotide scan.

Due to the short plasma half-life (less than 3 min) of somatostatin, metabolically stable analogs have been synthesized for clinical application. The first-generation somatostatin receptor ligands octreotide (OCT) and lanreotide (LAN) are stable octapeptides with a preferential binding affinity for sst_2_. A long-acting, repeatable octreotide formulation (octreotide acetate long-acting repeatable, LAR) consisting of octreotide incorporated into microspheres of the biodegradable polymer poly(DL-lactide-co-glycolide glucose) with preserved activity and a reduced number of administrations was developed and approved in 1995.

Somatostatin analogs have recognized activity in hormone-secreting pituitary adenomas and neuroendocrine tumors. However, since somatostatin analogs became available, the antiproliferative properties and tumor expression of ssts have continued to feed the idea of the extended therapeutic applications of SS analogs over the decades.

### 2.2. Somatostatin in the Thymus

In the late 1990s, interest grew in the role of the thymus as an endocrine gland [32]. Several hormones and neuropeptides were detected in thymic cells and specific receptors were found to be expressed on both thymic epithelial cells (TECs) and thymocytes, suggesting paracrine circuits that interfere with immune cell development and reception of signals coming from endocrine glands [33,34]. Moreover, immune cells were shown to express somatostatin and its receptors [33,35].

The first evidence of somatostatin gene expression in the rat thymus was provided in 1989 by Fuller and Verity, which suggested there was a possible paracrine role in modulating T-lymphocyte development [36]. Increasing proofs documented the synthesis of somatostatin in lymphoid organs, such as the thymus and the spleen [37].

Cortical and medullary TECs expressed high levels of somatostatin, while sst_2_ was expressed on thymocytes. In fetal thymic organ culture, somatostatin was found to enhance thymocytes’ proliferation and maturation [38] through sst_1_ and sst_2_ signaling [39]. Somatostatin and octreotide were shown to modulate the thymocytes’ development and maturation [40,41], which is essential in T cell selection and the induction of tolerance.

In vivo studies showed considerable somatostatin receptor expression by thymic tumors before the in vitro characterization. A high uptake of indium-labeled octreotide (^111^In-DTPA-d-Phe¹-octreotide) was shown in a series of 13 thymic tumors but not in histologically diagnosed benign thymic hyperplasia [42,43].

Intensive research from the Netherlands has detailed somatostatin receptor expression both in the normal and neoplastic thymus [33,44]. A normal human thymus expresses somatostatin (SS) mRNA and sst_1_, sst_2A_, and sst_3_, while cultured TECs were found to selectively express sst_1_ and sst_2A,_ and their proliferation was inhibited by somatostatin and octreotide [44]. Thereafter, a case of thymoma was studied by in vivo and in vitro techniques. In this case, sst_3_ was found to have a predominant role, with an uncontrolled growth supposed to be related to the absence of somatostatin [45].

Immunohistochemistry documented somatostatin receptor expression in approximately 80% of a series of thymic tumors studied by Ferone et al. [46]. The staining was highly heterogeneous, with a differential expression of the receptor subtype depending on the cell: sst_3_ expression was predominantly associated with thymocytes, and sst_2A_ expression was confined to malignant epithelial cells or within stromal structures.

## 3. Somatostatin Analogs in Clinics

The concept of somatostatin receptor imaging and therapy for tumors showing uptake has rapidly gained acceptance and can be considered one of the first targeted treatments.

### 3.1. Single-Case Reports

Significant tumor shrinkage was reported in a patient with a malignant thymoma and pure red cell anemia, which was treated with octreotide (1.5 mg/day) plus prednisone (0.6 mg/kg/day). Along with tumor control, the anemia also improved, while the single drugs used alone were ineffective [47].

Thereafter, several single cases of successful treatment with octreotide and prednisone were reported.

The most recent report was published in 2018. A metastatic thymic carcinoid patient treated with chemotherapy, concurrent radiotherapy, and octreotide, and, thereafter, with octreotide for maintenance therapy, reached an 18-month PFS [48].

A pretreated patient reportedly showed a good clinical and radiological response to third-line palliative octreotide plus steroid therapy with a positive octreotide scan [49].

A patient with a refractory recurrent B3 thymoma reportedly displayed tumor reduction under octreotide and steroid treatment [50].

A patient with a malignant B2 heavily pretreated thymoma reportedly showed a significant response on metastatic sites and stable disease on primary with octreotide and prednisone-based therapy, even though the preliminary octreotide scan revealed low uptake [51].

A response related to tumor and pure red cell aplasia was reported with octreotide alone in a pretreated patient showing only weak tracer uptake on octreotide scintigraphy [52].

A pediatric case with refractory thymic carcinoma with the expression of somatostatin receptors documented by octreotide scintigraphy underwent octreotide-plus-steroid treatment with a significant tumor response, which lasted only six months. At this progression, sites of metastases were found to be ssts negative [53].

### 3.2. Case Series

After a report in NEJM [47], a phase II study was carried out on 16 patients with advanced thymic tumors that were unresponsive to conventional chemotherapeutic regimens and were treated with octreotide (1.5 mg/day) plus prednisone (0.6 mg/kg/day, reduced to 0.2 mg/kg/day orally after the first three months) within the same institution [53]. In 8 cases, octreotide was replaced by the long-acting analog lanreotide (30 mg every 14 days intramuscularly) for a more practical schedule. The overall response rate reported was 37%, with 1 complete response and 5 (31%) partial responses. The disease control rate—which included the objective responses plus disease stabilization—achieved was 75%. The median survival was 15 months and the median time to progression was 14 months after a median follow-up of 43 months. Treatment was generally well tolerated, with acceptable toxicity. These results were considered clinically significant given that all enrolled patients had progressive and symptomatic disease at the beginning of therapy.

Figure 1 reports 1 case of our series. The patient was pretreated with chemotherapy and, at relapse, showed a positive 68Ga-DOTATOC (68Ga-Edotreotide) PET.

Given the significant results reported in this series, an ECOG multicenter study was started to define the role of octreotide and prednisone in advanced thymic tumors, including one thymic NET [54]. The treatment schedule contemplated an induction phase with octreotide alone, followed by the same treatment if an objective response was recorded, or followed by supplementation with prednisone if the disease was stable, while a progression determined study exit. Octreotide was administered at a dose of 0.5 mg subcutaneously 3 times a day for a maximum of 1 year and prednisone at a dose of 0.6 mg/day. Among 38 assessable patients, 4 partial responses were reported with octreotide alone (4/38, 10.5%), and 2 complete responses (2/38, 5%) and 6 partial responses (8/38, 21%) with octreotide associated with prednisone. The overall objective response rate reached 31.6% and included 2 complete and 10 partial responses. The median PFS (mPFS) for octreotide alone was 2 months and that of the combination of octreotide and prednisone was 9.2 months. However, we highlight that the confidence interval of mPFS for the octreotide group was particularly wide, ranging from 1.8 to 11.0 months, and was conditioned by study exit at progression after 2 months of therapy. The increased aggressive behavior of thymic carcinoma was reflected in the PFS of 4.5 months as compared to a PFS of 8.8 months for thymomas. The 1- and 2-year survival rates were 86.6% and 75.7%, respectively.

As the authors recognize, the study design does not allow for definitive conclusions on the activity of octreotide alone. Some later responses were reported in patients exhibiting minor responses at the 2-month evaluation. Somatostatin analogs more frequently induce the slow shrinkage of tumor and disease stabilization. Therefore, the short-term induction with octreotide may be questionable. Lethal toxicity secondary to a grade 5 infection without neutropenia was reported. Details about the timing of the event and the immune status of the patient were lacking in the study by Loehrer et al. [54]. The relationship with treatment was not clearly established, given the frequent association of immune disturbances in thymoma [55,56,57,58,59]. The hematologic effects (anemia, leukopenia, thrombocytopenia, etc.) reported were also unlikely to be determined by octreotide, as the safety data on somatostatin analogs confirm [60,61]. The eligibility criteria of the ECOG study harbor another potential confounding factor: all patients receiving corticosteroids for myasthenia gravis received the same dosage after entering the study. This means that the myasthenic patients were administered steroids also during the two months they were supposed to be on octreotide alone. The prolonged use of a rather high dose of prednisone used in the ECOG study may also have favored the predisposition to infection.

A neoadjuvant study from a German group showed the activity of octreotide LAR (30 mg once every 2 weeks) in combination with prednisone (0.6 mg/kg/day) administered for a maximum of 24 weeks in a series of 17 patients with unresectable TET [62]. The objective response rate was 88%. In this report, the response evaluation was performed at Week 12, supporting the need for a definite time of therapy to induce a response. One patient died from pulmonary sepsis during the study. However, this event was not related to the study medication but to the thymoma-associated immunodeficiency. In addition, other reported adverse events, such as deep-vein thrombosis, pulmonary embolism, pneumonia, rectal ulcer, sepsis, and necrotizing fasciitis, were clearly defined as unrelated to the study treatment.

In Table 1, a comprehensive view of the single-case reports and case series is provided.

## 4. Steroids in TET: Essential or Ancillary Drugs?

The thymus was defined as an endocrine gland because of the expression of various classical hormones and related receptors, such as corticotropin-releasing factor, adrenocorticotropic hormone (ACTH), ACTH receptors, and glucocorticoids. Thymus epithelial cells and thymocytes have an autonomous system of steroidogenesis, starting from the resource of the endogenous substrate, which is available in the thymus [71].

Studies on the effect of steroids on lymphocytes are rather old. Both T- and B-lymphocyte counts were changed by prednisone with a marked reduction in blood T cells [72].

Two opposite effects take place in the thymus in the absence or excess of steroids. The first instance is represented by Addison’s disease and adrenalectomized rats, where enlargement of the thymus gland was recognized [73,74]. On the contrary, steroid treatment determines the depletion of thymic lymphoid elements with increasing fat and connective tissue, as observed in the thymus from myasthenic patients. This change is usually representative of an atrophy process.

Glucocorticoid receptors are expressed by cultured TECs [75], and glucocorticoid hormones regulate thymic epithelial cell proliferation [76].

As usual, there are various reciprocal interferences between hormones. For example, thymulin and thymopentin, two typical thymic hormones, enhance the levels of ACTH [77], while glucocorticoid hormones influence the production of thymic hormones [76]. ACTH and glucocorticoids act in the same direction in regard to thymocytes: ACTH inhibits the mitogenesis of immature and mature thymocytes while glucocorticoids induce apoptosis of T cells and mostly of immature thymocytes [78,79].

Following the investigations on the normal thymus, the activity of steroids was studied in thymic tumors. The lymphocytic effect was confirmed and unrelated to the thymoma cell type [80,81].

Glucocorticoid receptor (GR) expression on neoplastic TECs and the effect of glucocorticoids on the cell cycle of cancer cells were studied in vitro [82]. GRs were shown to be expressed on neoplastic TECs as well as on non-neoplastic thymocytes in thymomas, regardless of WHO histological classification. Glucocorticoids induced the arrest of TECs in the G0/G1 phase in six cases examined and promoted apoptosis in the three cases with the lowest levels of Bcl-2 expression.

The use of steroids for thymoma-associated paraneoplastic syndrome and, among them, most frequently myasthenia leads to the clinical response to the tumor. There are several reports on the effective treatment of thymic tumors with steroids [83,84,85,86,87,88,89,90,91]. Thymic tumors that are refractory to chemotherapy have shown an objective response to steroid therapy [86,87]. In some of these case series, high doses of corticosteroids were used [88,89,90]. Hayashi et al. reported the clinical response obtained in an invasive thymoma with a platinum-based regimen plus a high dose of methylprednisolone (1000 mg on days 1–5 and 500 mg on days 6 and 7) for three courses [89]. This regimen was confirmed to be effective in a series of 14 patients with advanced invasive thymoma [90].

Glucocorticoids were administered as a neoadjuvant treatment in a series of 17 untreated patients with invasive thymoma [91]. The steroid pulse consisted of 1 g of methylprednisone each day for 3 days. The overall response rate reported was 47.1% (8 of 17). A significant reduction in the population of double-positive CD4+8+ immature thymocytes expressing high levels of glucocorticoids receptor was shown, which led to the tumor shrinkage that was predominantly obtained in B1 thymomas. Apoptotic changes were observed not only in lymphocytes but also in neoplastic epithelial cells.

In some cases, the need for the prolonged use of steroids was highlighted [45,46,65,76]. Steroids may also produce a response in thymoma that are refractory to octreotide [67].

Recently, a pretreated thymoma with an associated multi-organ autoimmunity showed a partial response to low-dose prednisone (0.5 mg/kg/day) [92].

## 5. Octreotide Plus Prednisone: Are Two Better than One?

A further question remains unsolved. Few data are available on the reciprocal interferences between somatostatin and prednisone.

The expression of ssts subtypes can be differentially regulated by GC depending on the dose used, the duration of treatment, and the tissue type examined [93]. Dexamethasone inhibits somatostatin production in a human medullary thyroid carcinoma cell line in a dose-related manner [94]. Preclinical studies have shown that the levels of thymic somatostatin mRNA exhibit a bell-shaped response to dexamethasone administration [36]. Low doses of dexamethasone enhance the expression of the somatostatin gene in the thymic gland [20]. In clinical therapeutic use, steroids are frequently used because of their associated thymoma autoimmune disorders. Even in the previously cited ECOG study that imposed a phase of octreotide only, myasthenic patients were allowed to use steroids [54].

The role of each drug in the treatment of TETs will remain an obscure issue.

Today, we have again come to rely on the considerations initially made in NEJM [47]: the apoptotic effects of corticosteroids exerted on the lymphocytes joined with the growth-inhibiting action of octreotide on the neoplastic thymic cells may at least partially explain the activity of the combination. The case report directed our choice to continue the combined use even if the first patient had pure red cell anemia, and discrimination among the patients with and without paraneoplastic syndromes should be performed.

## 6. Discussion

Octreotide has recently celebrated its 40th anniversary [95]. While its role is confirmed in acromegaly and neuroendocrine tumors, the research concerning somatostatin, the thymus, and thymic tumors seems to be exhausted. As an example, after the paramount results of the research by Lamberts and Hofland in the Netherlands in the first decade of 2000, this workstream has come to a halt. Given the historical background of our group, we wonder whether there is still a role for somatostatin analogs in TET.

Thymic neuroendocrine tumors (TNETs) may be a candidate for somatostatin analogs. They are aggressive, frequently metastatic tumors with a low (5 year) survival rate (~50%). They are typically symptomatic and most often secrete ACTH (40% of tumors) [96]. The European Society of Medical Oncology (ESMO) Clinical Practice Guidelines for Lung and Thymic Carcinoids recommends somatostatin analogs as the first-line therapy in typical carcinoids [97]. Regarding TETs, the National Comprehensive Cancer Network (NCCN) guidelines [1], the Esmo Clinical Guidelines since 2015 [98], the RYTHMIC Network [99], and recently the Italian Association of Medical Oncology [100] continue to consider this therapy among those recommended in the second-line setting. It is difficult to suppose that new multicenter studies will be designed to definitively assess the role of somatostatin analogs plus prednisone in TETs. Our feeling is that this therapy should be considered in tumors showing in vivo uptake of labeled octreotide, which remains an easy way to identify a candidate for potential effective treatment. Moreover, this association might be considered in less aggressive tumors enriched in lymphocytes, such as thymomas more than thymic carcinomas—even if occasional responses are reported also in this subtype. Thymic carcinomas treated with somatostatin analogs represent a niche. Therefore, we can only formulate suggestions for lack of compelling evidence.

However, a new landscape can be drawn.

TETs are theoretically not the ideal candidate for immune checkpoint inhibitors because of the increased risk of immune-adverse events and the lowest TMB among all cancers in adults. However, they show high expression of PD-L1 on tumor cells and abundant CD8+ lymphocytes that support the clinical use of ICIs [9]. Ongoing clinical studies will define the role of this therapy.

The other candidate is targeted therapy. The Akt/mTOR pathway is one of the most involved pathways in cell proliferation associated with cancer. Different from the normal thymus, both Akt and mTOR were activated in thymomas, and rapamycin, a specific inhibitor of mTOR, was shown to significantly reduce the proliferation of thymoma-derived epithelial cells [101]. In 2012, we reported on the response to everolimus achieved in two heavily pretreated patients with advanced TETs [102]. This preliminary evidence was confirmed in a phase II study from the French RYTHMIC network enrolling 51 patients, which reported a disease control rate as high as 88% [17]. The search for an actionable mutation in thymoma led to the treatment of a patient with a pathogenetic BRCA mutation with the PARP inhibitor olaparib, which achieved a significant clinical benefit [103].

Given the somatostatin receptor expression by TETs, the third definable chance is treatment with peptide receptor radionuclide therapy (PRRT) with Lu DOTATATE (Lutathera^®^). Lutathera^®^ is the first radiopharmaceutical for PRRT approved by the EMA in 2017 and the FDA in 2018 for the treatment of SSTR-positive gastroenteropancreatic neuroendocrine tumors [104]. In 2021, the Japanese Pharmaceuticals and Medical Devices Agency approved Lutathera^®^ for the treatment of somatostatin-receptor-positive neuroendocrine tumors. Lutathera^®^ is formed by the somatostatin analog DOTA-TATE combined with radionuclide 177Lu, thus delivering ionizing radiation specifically to somatostatin-receptor-expressing tumor cells. As a result, DNA single- and double-strand breaks are provoked, inducing cell death of the tumor and metastatic sites expressing ssts. One case treatment with Lutathera^®^ is reported [105].

The new frontier is represented by the combination of 177Lu-DOTA-TATE and targeted therapies such as olaparib in somatostatin-receptor-positive tumors (LuPARP trial NCT04375267) [106]. This is a phase I recruiting trial.

## 7. Conclusions

Preclinical and clinical investigations concerning octreotide or prednisone, or a combination of them, seem currently on hold. This field, as well as the possible synergism between these therapeutical agents, remains open.

The present study deals with our own experience with octreotide and at the same time reviews the literature on this complex subject. The current guidelines acknowledge the role of somatostatin in TETs. This treatment can be considered in pretreated patients showing in vivo expression of somatostatin receptors. Due to the reduced side effects, it is an advisable choice for patients with poor performance status. Beyond this, we can suppose an increase in related studies in the future, including studies on ICIs or drugs interfering with key cell signaling. Cooperative studies are needed to address the issues related to these rare tumors, along with convergent experience of high volume and experience centers.

## Figures and Tables

**Figure 1 cancers-14-00774-f001:**
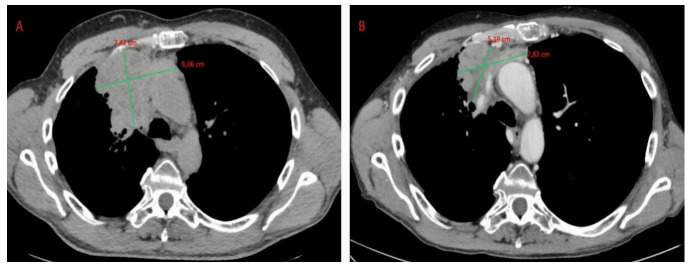
CT scan showing an anterior mediastinal mass. (**A**) Before and (**B**) after treatment with octreotide and prednisone. The mass appeared significantly reduced.

**Table 1 cancers-14-00774-t001:** Summary of published data on octreotide in thymoma from 1997 until today: TTP: time to progression; CR: complete response; PR: partial response; SD: stable response; PD: progressive disease; SRS: somatostatin receptor scintigraphy.

First Author	Year	Patients	Stage	Histotype	Responses	TTP	OS
Palmieri [47]	1997	1	-	-	1 CR	-	-
Lyn [63]	1999	1	-	-	1 CR	-	-
Palmieri [64]	2002	16	II, III, IV A/B	A-C	1 CR 5 PR 6 SD 4 PD	15 months (CI 12–28)	22.5 months (CI 7–43)
Loehrer [54]	2004	38	III–IV	-	2 CR 10 PR 14 SD 12 PD	8.8 months (CI 3, 7–12, 3)	9.2 months (CI 8, 1–13, 9)
Rosati [65]	2005	8 patients with positive SRS	I, II, III, IV A/B	-	7 SD	-	-
Longo [66]	2005	29; 7 patients with positive SRS	I, II, III, IV A/B	-	7 SD	-	-
Tiseo [67]	2005	1	III	B3	CR	-	-
Zaucha [52]	2007	1	-	-	PR	>12 months	-
Ito [50]	2009	1	III	B3	PR	-	-
Pettit [49]	2011	1	I	-	CR	-	-
Longo [68]	2012	44; 12 patients with positive SRS	I, II, III, IV	A, B2, B3, C	3 PR 5 SD 4 PD	8 months	-
Kirzinger [62]	2016	17	III	A, AB, B1, B2, B3, C	ORR 88%	-	-
Ottaviano [69]	2017	26	-	-	ORR 42%	21 months	88 months
Rutkowska [70]	2019	1	III	B2	CR	-	-
Sorejs [51]	2020	1	IV	B2	PR	8 months	-

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
