# Peer review of "The Never-Ending History of Octreotide in Thymic Tumors: A Vintage or A Contemporary Drug?"

_cancers, 2022, doi:10.3390/cancers14030774_

Round 1

Reviewer 1 Report

This review article entitled “The never-ending history of octreotide in thymic tumors: a vintage or contemporary drug? (cancers-1522831)” by Dr. Montella reviewed the treatment entities of thymic neuroendocrine tumors and other thymic malignancies focused on octreotide.

I felt this review article was comprehensive from old history to the new treatment, biological to clinical issues, and case to a prospective study in thymic malignancies concerning somatostatin analogue and meaning of prednisolone. I felt some structure issues were present. However, this review article will be of interest to readers for a thoracic oncologist.

Major.

I recommend the authors to add the clinical details in the European countries.

Minor.

Page 1 L26 I think thymic malignancies are basically located in the “anterior” mediastinum.

Page 1-L20 advanced chemo-refractory tumors~; I did not understand this sentence.

P1-L23 “larger studies” that the authors defined were kinds of the single arm with a small sample size for me.

P2 L 84 I recommend the authors add the recent phase 2 studies related to multikinase inhibitors as c-kit inhibitors.

P4 L172 “at the end of nineties” means 1990’s?

In the ECOG study, only one thymic NET patient was included. Please add it adequately.

Discussion

L 404. Lutathera is also approved by PMDA in Japan in 2021.

Author Response

Dear reviewer,

Thank you for the favorable comments. Enclosed you will find the revised version of paper no.1522831 and the following point-by-point reply.

Major.

I recommend the authors to add the clinical details in the European countries.

  • On page 9, line 375 we included the Esmo guidelines and the Rythmic network that cite the somatostatin treatment and relative references (99-100). The position paper concerning thymic carcinoids was already present mentioned in the text (reference 97)

 Minor.

Page 1 L26 I think thymic malignancies are basically located in the “anterior” mediastinum.

  • We have specified the location of thymic malignancies accordingly

Page 1-L20 advanced chemo-refractory tumors~; I did not understand this sentence.

  • We refer to surgically unresectable advanced and metastatic tumors which often do not exhibit a response to first-line platinum-based therapy. The sentence has been changed

P1-L23 “larger studies” that the authors defined were kinds of the single arm with a small sample size for me.

  • You are right, however, larger was used in comparison to single case reports. Surely this adjective may be confusing, so we have eliminated changing it into “single-arm phase II studies”

P2 L 84 I recommend the authors add the recent phase 2 studies related to multikinase inhibitors as c-kit inhibitors.

  • Thank you very much for your suggestion. We have included the Remora trial corresponding to reference 19

P4 L172 “at the end of nineties” means 1990’s?

  • We have modified it as follows: “in the late ‘90s”.

 In the ECOG study, only one thymic NET patient was included. Please add it adequately.

  • Page 6, Line 327: we have specified that one patient with thymic NET was included in the ECOG study

 Discussion

L 404. Lutathera is also approved by PMDA in Japan in 2021.

  • We included the note concerning Lutathera's approval in Japan.

Reviewer 2 Report

Thymic Epithelial Tumors are rare tumors frequently associated with paraneoplastic syndromes, Advanced chemo-refractory tumors represent a medical challenge and somatostatin receptor expression was documented in thymic tumors and represents the rationale for therapeutic use. However the role of these drugs in thymic epithelial tumors is still rather undefined.

This study deals with their own experience with octreotide and reviewed the literature on this complex subject. The role of somatostatin in TETs is acknowledged by the current guidelines and advisable for patients with poor performance status due to its reducing side effects. However cooperative studies are needed to address the issues related to TETs with convergent experience of high volume and experience centers.

Author Response

We are grateful to you for the favorable opinion expressed.

We agreed with the need for cooperative studies promoted by high-volume and experience groups.

Reviewer 3 Report

The never-ending history of ocreotide in thymic tumors: a vintage or contemporary drug?

I think this is very important study.

The recurrence of thymoma and invasive thymoma don’t have the suitable therapy for along time. Thymic Epithelial Tumors (TETs) are located at the difficult anatomical area for surgery.

Therefore we have to investigate the suitable therapy for TETs.

This is a premature report. This is a review study, and authors experiences have a little bit.

Almost all the data is the review of the article.

Authors have to investigate more precise basic research for somatostatin therapy.

  1. OS and DFS for octreotide therapy are unclear. Do authors have any data for OS and DFS for octreotide therapy ?
  2. Basically, surgical resection is the effective therapy for TETs

Do authors have any data for surgical therapy related to octreotide therapy ?

  1. Steroid has a large important part for the thymoma therapy.

How do authors think about the steroid role about the effectiveness of the combination therapy (octreotide and steroid ) ?

Author Response

We thank you for the favorable opinion expressed.

We agreed with you as concerns the limits of the study: it is only a review based on our experience and the available data, which are comprehensively scarce both as regards the preclinical and clinical point of view.

This review aims to light up the interest in preclinical basic research whose results were reported in the manuscript but do not solve the doubts about the presumptive mechanisms of action.

  1. OS and DFS for octreotide therapy are unclear. Do authors have any data for OS and DFS for octreotide therapy?

- The reported studies on refractory advanced thymic tumors were both phase II studies and thereafter aimed to activity more than efficacy parameters. Given this premise, we integrated the reported data with OS and DFS for the ECOG study.

  • on page 5, lines 305-307 data related to the Cancer series were reported.
  • on page 7, lines 339-346 we detailed the data reported on OS and DFS from the ECOG study.
  1. Surgical resection is the effective therapy for TET. Do authors have any data for surgical therapy related to octreotide therapy?
  • Unfortunately, our treated advanced refractory patients did not become susceptible to surgery during this phase of the disease. This part warrants attention in further studies.
  1. Steroid has a large important part for the thymoma therapy. How do authors think about the steroid role about the effectiveness of the combination therapy (octreotide and steroid ) ?
  • We have included some considerations on page 9 lines 505-511.